

# Spatio-temporal modeling of air pollutant concentrations in Germany using machine learning

Vigneshkumar Balamurugan[1], Jia Chen[1], Adrian Wenzel[1], and Frank N. Keutsch[2,3]

[1]Environmental Sensing and Modeling, Technical University of Munich (TUM), Munich, Germany.
[2]School of Engineering and Applied Science, Harvard University, Cambridge, MA, USA.
[3]Department of Chemistry and Chemical Biology, Harvard University, Cambridge, MA, USA.

**Correspondence:** Vigneshkumar Balamurugan (vigneshkumar.balamurugan@tum.de), Jia Chen (jia.chen@tum.de)

**Abstract.** Machine learning (ML) models are becoming a meaningful tool for modeling air pollutant concentrations. ML models are capable of learning and modeling complex non-linear interactions between variables, and they require less computational effort than chemical transport models (CTMs). In this study, we used gradient boosted tree (GBT) and multi-layer perceptron (MLP; neural network) algorithms to model near-surface nitrogen dioxide ($NO_2$) and ozone ($O_3$) concentrations over Germany at 0.1 degree spatial resolution and daily intervals.

We trained the ML models using TROPOMI satellite column measurements combined with information on emission sources, air pollutant precursors and meteorology as feature variables. We found that the trained GBT model for $NO_2$ and $O_3$ explained a major portion of the observed concentrations ($R^2$ = 0.68-0.88, RMSE = 4.77-8.67 $\mu$g m$^{-3}$ and $R^2$ = 0.74-0.92, RMSE = 8.53-13.2 $\mu$g m$^{-3}$, respectively). The trained MLP model performed worse than the trained GBT model for both $NO_2$ and $O_3$

($R^2$ = 0.46-0.82 and $R^2$ = 0.42-0.9, respectively).

Our $NO_2$ GBT model outperforms the CAMS model, a data-assimilated CTM, but slightly under-performs for $O_3$. However, our $NO_2$ and $O_3$ ML models require less computational effort than CTM. Therefore, we can analyze people's exposure to near-surface $NO_2$ and $O_3$ with significantly less effort. During the study period (2018-04-30 and 2021-07-01), it was found that around 36% of people lived in locations where the WHO $NO_2$ limit was exceeded for more than 25% of the days, while 90%

of the population resided in areas where the WHO $O_3$ limit was surpassed for over 25% of days. Although metropolitan areas had high $NO_2$ concentrations, rural areas, particularly in southern Germany, had high $O_3$ concentrations.

Furthermore, our ML models can be used to evaluate the effectiveness of mitigation policies. Near-surface $NO_2$ and $O_3$ concentrations changes during the 2020 COVID-19 lockdown period over Germany were indeed reproduced by the GBT model, with meteorology-accounted for near-surface $NO_2$ significantly decreased (by 23±5.3%) and meteorology-accounted

for near-surface $O_3$ slightly increased (by 1±4.6%) over ten major German metropolitan areas, compared to 2019. Finally, our $O_3$ GBT model is highly transferable to other countries, at least to neighboring countries and locations where no measurements are available ($R^2$ = 0.87-0.94), whereas our $NO_2$ GBT model is moderately transferable ($R^2$ = 0.32-0.64).



## 1 Introduction

Air pollution is a major threat to human health and impacts ecosystems (Bell et al., 2011; Lelieveld et al., 2015; Zhang et al.,
2019; Xie et al., 2019). Based on the source, air pollutants are classified as primary (directly emitted from anthropogenic/-
natural sources) or secondary (formed through complex atmospheric chemical reactions). Near-surface nitrogen oxide ($NO_X$ =
NO+$NO_2$) is a primary air pollutant emitted largely by fossil-fuel-consuming sectors such as vehicles, industries, power plants,
etc., but there are also natural sources such as lightning, soil emissions, and biomass burning. Near-surface ozone ($O_3$) is a
secondary air pollutant produced solely by the photolysis of $NO_2$ (nitrogen dioxide) in the presence of sunlight (Crutzen, 1988;
Council et al., 1992).

Tropospheric $NO_X$ and $O_3$ are chemically strongly coupled via complex atmospheric chemical reactions (Jacob, 1999). The
majority of $NO_X$, from primary sources such as fossil-fuel combustion, is emitted in the form of nitric oxide (NO), which
rapidly converts to $NO_2$ by reacting with $O_3$. In turn, $O_3$ and NO are generated again by photolysis of $NO_2$, forming a null
cycle. Therefore, the amount of sunlight present and the total concentration of $NO_X$ determine ozone production via this
$NO_X$ null cycle. However, the oxidation of volatile organic compounds (VOCs) via the hydroxyl (OH) radical results in the
formation of hydro-peroxy radicals ($HO_2$) and organic-peroxy radicals ($RO_2$), which can alter the NO/$NO_2$ ratio. The presence
of hydroxyl radical initiates the VOC oxidation process, followed by the formation of hydro- and organic peroxy radicals,
which convert the NO to $NO_2$, which can form additional $O_3$, as well as converting $HO_2$ back to OH thus forming a catalytic
cycle ($HO_X$ catalytic cycle). However, ozone production is non-linear in relation to its precursors ($NO_X$ and VOC) due to
termination reactions that occur within the catalytic cycle (Lin et al., 1988; Nussbaumer and Cohen, 2020; Pusede and Cohen,
2012; Pusede et al., 2014). To that end, the response of ozone production is categorized into three regimes: $NO_X$-saturated
(high $NO_X$ with low VOC), $NO_X$-limited (low $NO_X$ with high VOC), and transitional (Sillman et al., 1990; Sillman, 1999). In
the $NO_X$-saturated regime (typically urban areas), ozone production is inversely proportional to $NO_X$ concentration, whereas
ozone production is directly proportional to VOC concentration. However, in $NO_X$-limited regimes (typically rural areas),
ozone production is directly proportional to $NO_X$ concentration, whereas VOCs have little effect on ozone production. This
complex ozone production vs. precursor emission response is also evident in real-time observations, such as urban weekend
ozone levels being higher than weekday levels (Sicard et al., 2020) and high ozone levels during public holidays and national
shutdowns (e.g., the COVID-19 lockdown) due to low $NO_X$ emissions (Balamurugan et al., 2021, 2022b).

Chemical transport models (CTMs) are commonly used to study air pollution and its drivers (Hu et al., 2016; Lou et al.,
2015), but these models are dependent on emissions as represented in emission inventories (Pisoni et al., 2018). Emission
inventories are typically developed using the bottom-up method, based on data such as economic activity, fuel consumption,
traffic density, etc (McDuffie et al., 2020; Osses et al., 2022). However, bottom-up emission inventories can be highly uncertain
due to inaccuracies in the data used in the bottom-up method, especially from unaccounted sources (Chen et al., 2020; Crippa
et al., 2019; Trombetti et al., 2018). Because of the significant computational effort and storage space requirements, CTMs
often perform at coarse spatial resolution, making it unable to solve fine transport and chemical mechanisms, particularly over
complex topography (Singh et al., 2021). Machine learning (ML) models have been shown to be an effective complement to




these computationally expensive CTMs (Vlasenko et al., 2021). The performance of machine learning models for modeling air pollutants is promising (Balamurugan et al., 2022a; Cheng et al., 2022; Lee et al., 2020; Li et al., 2022; Liang et al., 2020; Liu et al., 2022; Zaini et al., 2022; Zhao et al., 2023). Meteorological variables such as solar radiation and temperature have been shown to be important parameters in near-surface ozone modeling using machine learning (Diao et al., 2021; Hu et al., 2021). Meteorological conditions influence the concentration of $O_3$ both directly and indirectly. Solar UV radiation is responsible for the photolysis of $O_3$ precursors ($NO_2$ and VOCs). Temperature directly influences the photochemical reaction rate. Furthermore, meteorology influences biogenic and fuel-leak-related VOC emissions (exponentially proportional to temperature), which account for a significant portion of total VOC emissions (Guenther et al., 1993). In addition to meteorology, when emission source information is included, ML models predict near-surface $NO_2$ very well (Ghahremanloo et al., 2021; De Hoogh et al., 2019).

In-situ air quality measurements are sparse and concentrated primarily in urban areas. Recent advancements in satellite remote sensing allow us to analyze urban and non-urban air quality with adequate spatial and temporal coverage; however, they typically only measure the total or tropospheric column of specific air quality species, making it difficult to interpret people's exposure to near-surface air pollutants concentration. Therefore, in this study, we trained two ML models for near-surface $NO_2$ and $O_3$ concentrations over Germany using available information on proxies for near-surface air pollutants (satellite column measurements) and emission sources, precursors of air pollutants, as well as meteorology. Many recent studies, similar to ours, have attempted to model near-surface $NO_2$ and $O_3$ concentrations at national/regional spans (De Hoogh et al., 2019; Kang et al., 2021; Li et al., 2020; Zhu et al., 2022); there are, however, very few attempts over Germany. To the best of the authors' knowledge, only one study (Chan et al., 2021) used TROPOMI satellite $NO_2$ tropospheric column measurements and other auxiliary information (e.g., meteorology) to model near-surface $NO_2$ concentrations over Germany using a MLP model. Furthermore, previous studies have focused on a single pollutant (e.g., $NO_2$), whereas in this study, we model and analyze the spatio-temporal variations in both $NO_2$ and $O_3$, which are chemically strongly coupled. In terms of anthropogenic emissions, we also evaluate the ML model performance of $NO_2$ and $O_3$ during the 2020 COVID-19 lockdown period, which serves as a natural experiment period with significantly lower primary anthropogenic emissions (Gensheimer et al., 2021).

## 2 Study region, Data sets, Model, and Method

All data sets used in this study, as well as their spatial and temporal resolutions, are summarized in Table 1.

### 2.1 Study region and near-surface $NO_2$ and $O_3$ measurements

We focused on the spatial domain of 5-15°E and 47-55.5°N, particularly over Germany. Near-surface $NO_2$ and $O_3$ data from measurement stations across Germany were used in this study. However, not all measuring stations collect data on both pollutants; there are less stations measuring $O_3$ than those measuring $NO_2$. There were also temporal gaps in the measurement data. Therefore, we only considered stations that had more than 80% data coverage during the study period. In the end, we





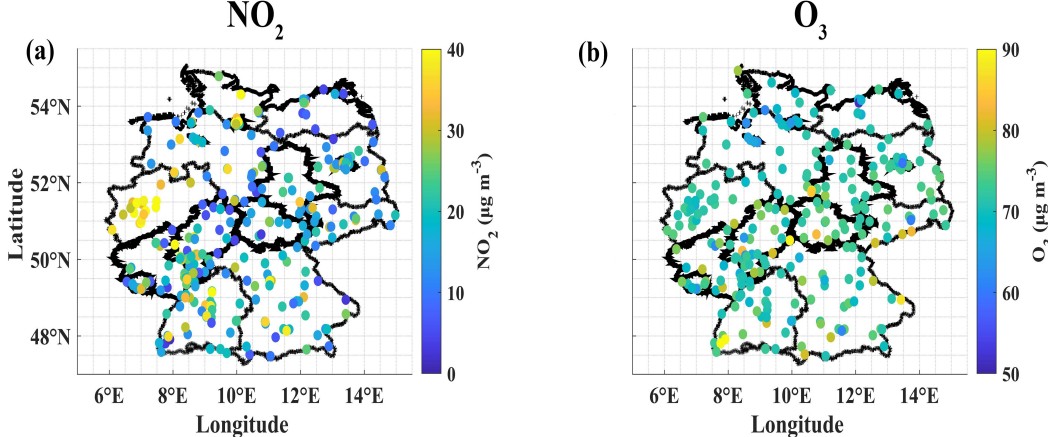

**Figure 1.** Locations of near-surface NO$_2$ (a) and O$_3$ (b) measurement stations considered in this study. The color bar depicts the mean of near-surface NO$_2$ and O$_3$ for each measurement station during the study period.

considered 321 stations for modeling NO$_2$ and 256 stations for modeling O$_3$. The selected measurement stations are located throughout the entire country and are situated in high-traffic, industrial, and background locations (Fig. 1 & Table A1).

## 2.2 Predictor variables of ML model

Predictor variables or input features for the ML models include satellite column measurements of air pollutants, meteorology and auxiliary data containing information on the area of interest.

### 2.2.1 Satellite column measurements

Tropospheric column NO$_2$, total column O$_3$, and troposheric column HCHO data are used, which are level-2 retrieval products from TROPOMI (TROPOspheric Monitoring Instrument) aboard the Sentinel-5P satellite. Sentinel-5P overpasses the study area between 13:00 and 14:00 local standard time. The spatial resolution of TROPOMI data is 7*3.5 km (increased to 5.5*3.5 km after August 6, 2019). We applied the data quality filtering described in the product manual to each data product (S5P (2022b) for NO$_2$, S5P (2022c) for O$_3$, and S5P (2022a) for HCHO). Tropospheric column NO$_2$ is used in the NO$_2$ ML model because it can be considered as proxy for near-surface NO$_2$. Since NO$_2$ is the precursor for O$_3$, we also included the tropospheric column NO$_2$ in the O$_3$ ML model. Because formaldehyde (HCHO) is an intermediate gas-product of VOC oxidation, it can be used as a proxy for VOC-oxidation (Jin et al., 2017). Therefore, we included tropospheric column HCHO in the O$_3$ model. We also considered the "TROPOMI FNR" (ratio of "TROPOMI HCHO" and "TROPOMI NO$_2$") in the O$_3$ ML model, which in previous studies has been shown to be a useful indicator of ozone production regime (Jin et al., 2020; Wang et al., 2021). We included total column O$_3$ in the O$_3$ ML model by considering total column O$_3$ as a proxy for near-surface O$_3$.





**Table 1.** Data sets and related information used in this study.

| Data source | Data | Temporal resolution | Spatial resolution |
|---|---|---|---|
| Governmental in situ measurements | Near-surface $NO_2$ and $O_3$ | 1 hr | - |
| TROPOMI satellite measurements | Tropospheric column $NO_2$, total column $O_3$ and tropospheric column HCHO | Daily | 7 km*3.5 km (5.5 km*3.5 km, after 6 August 2019) |
| ERA5 (ECMWF reanalysis) | Temperature, relative humidity, wind speed, wind direction, downwind UV solar radiation at surface, boundary layer height, surface pressure and temperature of air at 2m above the surface | 1 hr | 0.25*0.25-degree |
| U.S. Geological Survey | Surface elevation | - | 1*1-km |
| GRIP global roads database | Road density | - | 8*8-km |
| CAMS European air quality reanalyses | Near-surface $NO_2$ and $O_3$ | 1 hr | 0.1*0.1-degree |
| GEOS-Chem chemical transport model | Near-surface $NO_2$ and $O_3$ | 1 hr | 0.5*0.625-degree |

### 2.2.2 Vegetation index

Normalized difference vegetation index (NDVI) and enhanced vegetation index (EVI) data were obtained from MODIS (Moderate Resolution Imaging Spectroradiometer) measurements aboard the Terra and Aqua satellites. We used the "MOD13A2 (16-day 1-km) VI" data set, which contains NDVI and EVI data at 1 km spatial resolution and 16 day temporal resolution. To generate daily intervals, the NDVI and EVI data were linearly interpolated. We considered these vegetation indexes in the $O_3$ ML model because vegetation contributes a considerable amount of VOCs. We also considered these vegetation indexes in the $NO_2$ ML model as a supplementary information to check whether changes in vegetation cover has any implications on $NO_2$ concentration changes.

### 2.2.3 Meteorology

Meteorology has both direct and indirect effects (e.g., dispersion, photochemical reactions) on pollutant concentrations. Meteorological variables such as temperature (T), relative humidity (RH), wind speed (WS), and wind direction (WD) were obtained from the ERA-5 reanalysis product. These variables were derived from the lowest model level (1000 hPa) of the "ERA-5 hourly data on pressure levels" data set. Downward UV solar radiation at the surface (DUV), boundary layer height (BLH), surface pressure (SP) and temperature of the air at 2 m above the surface (T2m) were derived from the "ERA-5 hourly data on single levels" data set. These meteorological data have a spatial resolution of 0.25 degree and a temporal resolution of one hour. In both the $NO_2$ and $O_3$ ML models, we took all meteorology variables into account.

### 2.2.4 Proxy for $NO_X$ emission source

Because vehicle (transport sector) emissions are a significant source of $NO_X$ emissions, considering a proxy for vehicle emissions is crucial. Therefore, we used road density as a proxy for the source of $NO_X$ emissions. We are aware that traffic volume





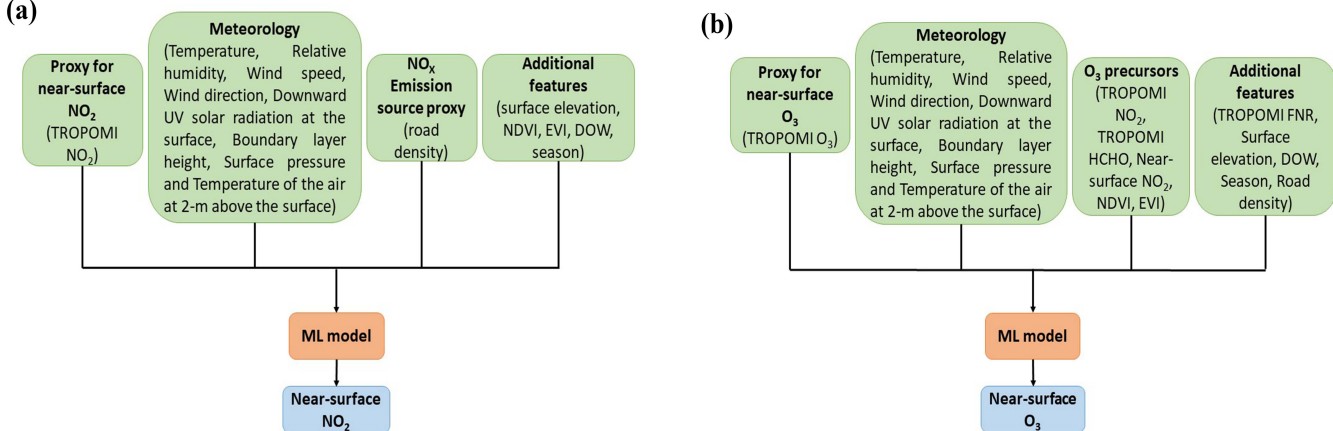

**Figure 2.** Predictor variables and data flow for the $NO_2$ (a) and $O_3$ (b) ML model.

or density would be the ideal proxy, but data on traffic volume or density on a national/regional span is not available. The road
density (RD) data was obtained from the GRIP global roads database, with a spatial resolution of 8 km.

### 2.2.5   Additional features

Additional supplementary data such as surface elevation (E) was obtained from the U.S. Geological Survey (USGS), with a
spatial resolution of 1 km. Surface elevation was taken into account because it influences the tropospheric/total column value
of measurements. We also considered "DOW" (day of the week), and "season" (season of the year) information in both the
$NO_2$ and $O_3$ models since both $NO_2$ and $O_3$ have distinct weekly and seasonal cycles. Because $NO_2$ is an important precursor
to $O_3$, in addition to "TROPOMI $NO_2$", we also included "Near-surface $NO_2$" modeled from $NO_2$ ML model as a feature
variable in the $O_3$ ML model.

### 2.3   Study period and data pre-processing

The study period was chosen to be between 2018-04-30 and 2021-07-01, which corresponds to the availability of TROPOMI
data retrievals with the same processing version. Despite the fact that satellites pass over the study area between 13:00 and 14:00
local standard time, we found that the satellite data represents the daily mean of air pollutants well. Therefore, we considered
the daily 24-hr mean for near-surface $NO_2$ and the daily maximum 8-hour mean (i.e. the mean of the 8 highest hourly values
during a day) for near-surface $O_3$ as our variables of interest (dependent variables to model), as these are commonly used
metrics in air quality research (Hoffmann et al., 2021).
Because each data set has a different spatial and temporal resolution, we re-sampled all of the data to the same spatial
(0.1*0.1 degree) and temporal (daily) resolution. The 0.1 degree ($\approx$ 10 km) resolution was chosen because it corresponds to
the resolution of the main features such as road density (spatial resolution of 8 km), TROPOMI satellite measurements (spatial



**Table 2.** Evaluation metrics of our GBT model in different testing strategies.

| | | Random (1-fold) | Random (5-fold) | Time-leave-out (5-fold) | Location-leave-out (5-fold) |
|---|---|---|---|---|---|
| **NO$_2$** | **R$^2$** | 0.88 | 0.89±0.002 | 0.74±0.07 | 0.68±0.12 |
| **GBT model** | **RMSE ($\mu$g m$^{-3}$)** | 4.77 | 4.65±0.034 | 6.77±0.7 | 8.67±1 |
| **O$_3$** | **R$^2$** | 0.92 | 0.92±0.001 | 0.74±0.09 | 0.8±0.06 |
| **GBT model** | **RMSE ($\mu$g m$^{-3}$)** | 8.53 | 9.36±0.068 | 13.2±1.1 | 12.45±1.3 |

resolution of 7*3.5 km), and concurrent high-resolution (0.1 degree) reanalysis air quality datasets from CAMS (Copernicus Atmosphere Monitoring Service). We computed the daily 24-hr mean for near-surface NO$_2$ and the daily maximum 8-hr mean
for near-surface O$_3$ for each in-situ measurement station and then calculated the mean of all stations that fell within 0.1 degree grid. The mean of surface elevation, NDVI, EVI, TROPOMI (NO$_2$, HCHO, O$_3$), and road density for each day were then calculated for the corresponding 0.1 degree grids. The surface elevation and road density were assumed to be constant during the study period. The ERA-5 meteorology product was resampled to 0.1 degree resolution using the nearest-neighbor method and the 24-hr mean was computed.

**2.4 Machine learning model and evaluation strategies**

We primarily used the gradient boosted tree (GBT) machine learning algorithm, XGBoost (Chen and Guestrin, 2016), to model near-surface NO$_2$ and O$_3$ concentrations. The GBT algorithm is a gradient-boosted decision tree-based algorithm that is expected to outperform deep neural network-based algorithms for structured data (Lundberg et al., 2020). Furthermore, tree-based models are more interpretable and require less time to train than deep neural network algorithms. However, for
comparison, we also used the multi-layer perceptron (MLP; neural network) algorithm (Gardner and Dorling, 1998). The GBT and MLP algorithms were implemented using "scikit-learn", a Python module (https://scikit-learn.org/stable/). When training the MLP model, we normalized the discrete feature variables between 0 and 1. The corresponding predictor variables and data flow for the NO$_2$ and O$_3$ ML model is shown in Fig. 2.

To evaluate the ML model, we used the R$^2$ (coefficient of determination) and RMSE (root-mean-square error) metrics. We
split the available data into training (70% of the data) and testing (the remaining 30%). The training data set was used to iteratively vary the hyper-parameters (combinations) and select the best set of hyper-parameters using a 5-fold CV (cross-validation). The hyper parameters used in this study are shown in Table A2 and Table A3. We also evaluated the ML model using three different 5-fold CV testing strategies (random 5-fold CV, time-leave-out 5-fold CV, and location-leave-out 5-fold CV) with 100% of the data (Meyer et al., 2018). In the random 5-fold CV testing strategy, the data was randomly split into
five parts, four of which were used for training and one for testing. This procedure was repeated until all five parts had been used as test. The mean (and standard deviation) of R$^2$ and RMSE from the 5-fold CV were then computed. In the time-leave-out 5-fold CV testing strategy, the 5-fold CV procedure was the same, but the data was split based on time period (by date;



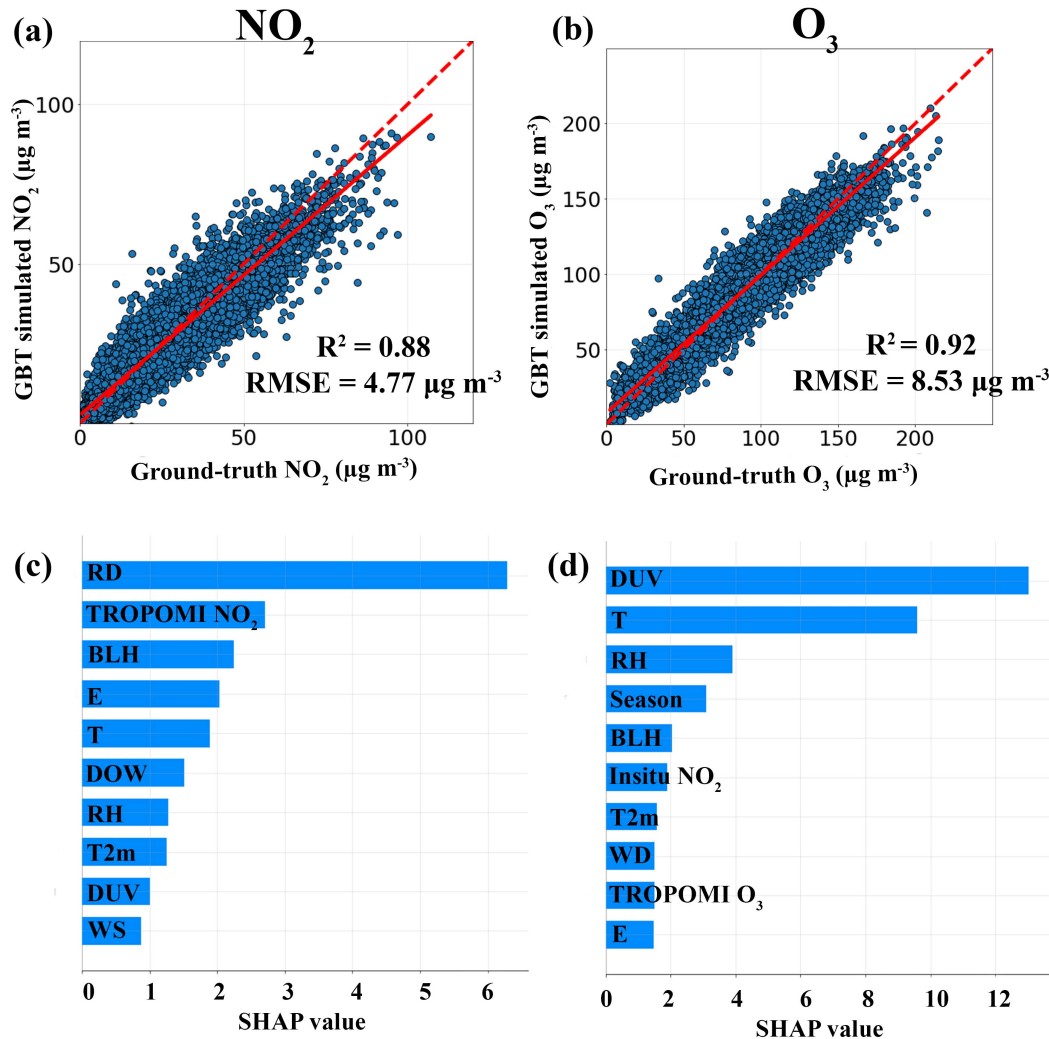

**Figure 3.** Comparison between ground-truth and GBT-simulated near-surface NO$_2$ (a) and O$_3$ (b). Feature importance (top 10) calculated based on SHAP (SHapley Additive exPlanations) values for NO$_2$ (c) and O$_3$ (d) GBT model. RD: Road Density, BLH: Boundary Layer Height, E: Surface Elevation, T-Temperature, DOW- Day of the week, RH-Relative Humidity, T2m: Temperature at 2 meter height, DUV: Downwind UV radiation, WS: Wind speed, WD: Wind Direction.





from the start of study period to the end of study period). Similarly, in the location-leave-out 5-fold CV testing strategy, the data was split based on location (by latitude). Figure A1 shows the first one-fold step in a 5-fold CV for time-leave-out and
location-leave-out testing strategies. To interpret the importance of feature variables in the fitted model, we use SHAP (SHapley Additive exPlanations) values. The SHAP method (https://christophm.github.io/interpretable-ml-book/shap.html) is the most commonly used method for interpreting ML model output, which calculates the contribution of each feature variable to the final prediction. Thus, higher SHAP values indicate greater feature importance.

## 2.5   CAMS model data

We obtained near-surface $NO_2$ and $O_3$ reanalysis data from CAMS in order to compare the performance of our ML model to that of the chemical transport model. This data set is based on a data-assimilation technique that combines real-time measurements with an ensemble of eleven air quality models to provide air quality data with high spatial resolution (0.1 degree) and 1 hr temporal resolution over Europe; however, it is only available for three years in the rolling archive. We used data from 2019-07-17 to 2020-01-31. We did not use data after 2020-01-31 due to COVID-19 lockdown restrictions, which limited many
anthropogenic emission activities, and CAMS had not adjusted the emission inventory for changes in emissions. Furthermore, because $NO_2$ has a shorter lifetime, the effect of assimilated observations is minimal, and the CAMS reanalysis $NO_2$ product mostly reflects emissions prescribed in the inventory (Inness et al., 2015).

## 2.6   GEOS-Chem model data

In this study, GEOS-Chem (GC) chemical transport model simulations were used to disentangle the meteorology contribution
when estimating the influence of COVID-19 lockdown restrictions on air pollutant concentration changes. The GC simulations over the study area were obtained with a spatial resolution of $0.5 \times 0.625$ degree and 1-hr temporal resolution for the 2020 strict COVID-19 lockdown period (March 21 to May 31) and the same period in 2019. Identical anthropogenic emissions from the 2014 CEDS inventory were used for both 2020 and 2019, but with the corresponding meteorology, natural, and fire emissions in the respective years. Therefore, the difference in GC-simulated species ($X$) concentrations between 2020 and
2019 results from changes in meteorology, natural, and fire emissions between 2020 and 2019 ($GC\ X_{2020-2019}$); here, $X$ refers to either $NO_2$ or $O_3$. Then, we subtracted the $GC\ X_{2020-2019}$ from the observed near-surface $X_{2020-2019}$ to estimate the changes in concentrations of species $X$ due to changes in anthropogenic emissions in the 2020 lockdown period (refer to studies Balamurugan et al. (2021); Qu et al. (2021) for the detailed description of the method).

## 3   Results

### 3.1   ML model evaluation and feature importance

The trained GBT model with 70% of the data for $NO_2$ reproduced the observed $NO_2$ concentration well in the test case, with an $R^2$ of 0.88 and RMSE of 4.77 $\mu$g m$^{-3}$ (Fig. 3(a) and Table 2). The random 5-fold CV results were in the same



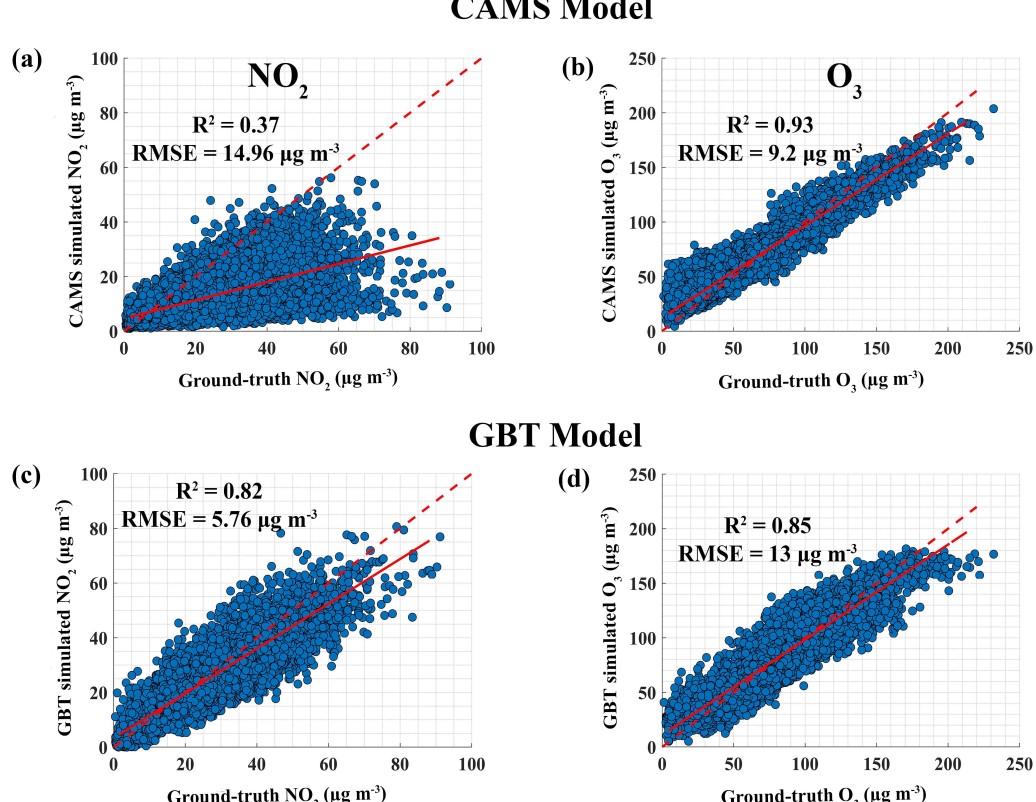

**Figure 4.** Top: Comparison between ground-truth near-surface NO$_2$ and CAMS reanalysis near-surface NO$_2$ (a) and O$_3$ (b) for the period between 17-07-2019 and 31-01-2020. Bottom: Comparison between ground-truth near-surface NO$_2$ and GBT-simulated near-surface NO$_2$ (c) and O$_3$ (d) for the period between 17-07-2019 and 31-01-2020. The dotted line represents a 1:1 line, while the solid line represents a linear fit.

range (R$^2$=0.89±0.002 and RMSE= 4.65±0.034 $\mu$g m$^{-3}$). The other two testing strategies (time-leave-out 5-fold CV and location-leave-out 5-fold CV) showed slightly worse agreement (Table 2), indicating that different validation strategies should
be performed to interpret the ML model capability. Otherwise, it may result in an overoptimistic view of ML models (Meyer et al., 2018). Furthermore, the worse agreement in the location-leave-out 5-fold CV testing strategy suggests that there is less confidence in modeling the near-surface NO$_2$ over new locations that the GBT model has not been trained on before. However, these results outperformed the MLP model trained by another study (Chan et al. (2021); R = 0.8 and RMSE = 6.32 $\mu$g m$^{-3}$ obtained for the testing strategy of random split of 90% of data used for training and 10% of data used for testing)
for near-surface NO$_2$ over Germany. Feature importance, based on the SHAP values, indicates that road density is the most important feature in the fitted model for NO$_2$ (Fig. 3(c)), because traffic is the main source of near-surface NO$_X$ in urban areas. The next most important features were TROPOMI NO$_2$, boundary layer height, and elevation. Because the majority of NO$_X$ sources are present at the surface, tropospheric column NO$_2$ data plays an important role in explaining near-surface





NO$_2$. Near-surface NO$_2$ typically has a negative correlation with boundary layer height, as increasing BLH disperses more and

vice versa (Balamurugan et al., 2021). Therefore, BLH is one of the most important features. It is unexpected that elevation was an important feature. The cause could be that the surface elevation varies greatly across Germany, influencing the total tropospheric column of NO$_2$ and thus serving as a link between the tropospheric column of NO$_2$ and near-surface NO$_2$. A previous study (Chan et al., 2021) also found that elevation was an important feature in the fitted MLP model for near-surface NO$_2$ over Germany.

The GBT model trained with 70% of the data for O$_3$ also well represented the observed O$_3$ concentrations in the test case, with an R$^2$ of 0.92 and RMSE of 8.53 $\mu$g m$^{-3}$ (Fig. 3(b)). Similar to the NO$_2$ GBT model findings, time-leave-out 5-fold CV and location-leave-out 5-fold CV testing strategies showed less agreement than the random 5-fold CV testing strategy (Table 2). In comparison to our NO$_2$ GBT model, our O$_3$ GBT model demonstrated greater confidence in modeling near-surface O$_3$ over locations the model was not trained on. According to SHAP values, the five most important features were DUV, T, RH,

BLH, and season, with DUV having the greatest influence (Fig. 3(d)). Because ozone is formed in the atmosphere from the photolysis of NO$_2$, DUV plays a significant role in the fitted model that explains near-surface O$_3$. Temperature is the second most important feature, which is also not surprising as it drives biogenic VOC emissions (an important precursor to O$_3$). Previous studies also show similar findings (Diao et al., 2021; Hu et al., 2021). GBT-modeled near-surface NO$_2$ was the sixth most important feature in the fitted model, according to the SHAP values, and it was also more important than TROPOMI

NO$_2$.

Figure A2 shows the results obtained from the MLP model. Both the NO$_2$ and O$_3$ MLP models performed worse than the NO$_2$ and O$_3$ GBT models, respectively (Table A4 vs. Table 2). In particular, MLP model findings showed low agreement in time-leave-out 5-fold CV and location-leave-out 5-fold CV testing strategies. This supports previous studies (Heaton, 2020; Lundberg et al., 2020) showing MLP model is unlikely to outperform tree-based models for tabular data. Because the GBT

model outperforms the MLP model, we only considered the GBT model results in the following.

## 3.2  GBT model performance compared to CAMS

To evaluate how well our GBT model performs compared to CAMS, we compared the high-resolution near-surface NO$_2$ and O$_3$ reanalysis data from CAMS with observations, and GBT-simulated near-surface NO$_2$ and O$_3$ with observations, for the period between 2019-07-17 and 2020-01-31, i.e., CAMS comparison period, (Fig. 4). Please note this time period was not used for

training the GBT model for this comparison. Our NO$_2$ GBT model reproduced the observed near-surface NO$_2$ concentrations well during this comparison period, with an R$^2$ of 0.82 and RMSE of 5.76 $\mu$g m$^{-3}$, while CAMS NO$_2$ reanalysis showed poor representation (R$^2$ = 0.37 and RMSE = 14.96 $\mu$g m$^{-3}$). However, CAMS O$_3$ reanalysis agreed slightly better with observed concentrations (R$^2$ = 0.93 and RMSE of 9.2 $\mu$g m$^{-3}$) compared to our O$_3$ GBT model (R$^2$ = 0.85 and RMSE = 13 $\mu$g m$^{-3}$). It should be noted that CAMS model reanalyses were based on data assimilation techniques. Therefore, the CAMS models are

expected to outperform our GBT models. However, our NO$_2$ GBT model outperforms CAMS, possibly because the effect of data assimilation is minimal in the CAMS reanalysis product due to the short NO$_2$ lifetime.



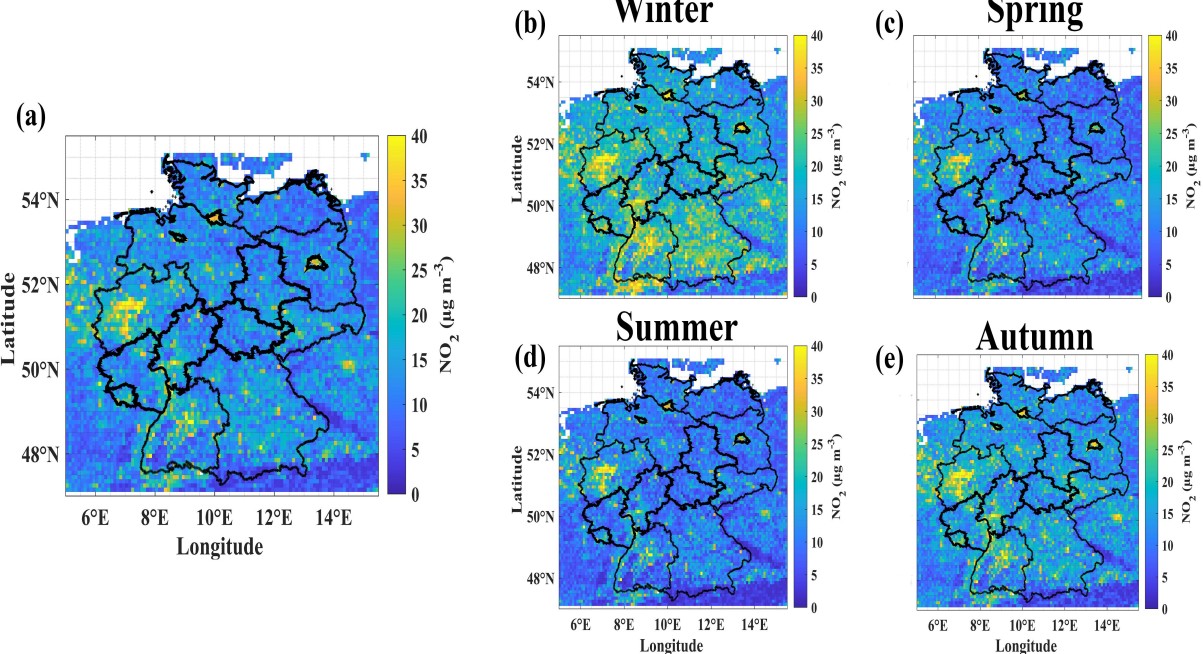

**Figure 5.** (a) Averaged GBT-simulated near-surface NO$_2$ concentrations over the study domain during the study period. (b-e) Averaged GBT-simulated near-surface NO$_2$ concentrations for each season during the study period.

### 3.3 Spatio-temporal changes in near-surface NO$_2$ and O$_3$ over the study domain

After the discussed model evaluation, we trained the GBT model using 100% of the data and modeled the near-surface NO$_2$ and O$_3$ concentrations over the study domain at 0.1 degree resolution and daily (24-hr mean for NO$_2$ and 8-hr maximum

mean for O$_3$) intervals. The averaged GBT-modeled near-surface NO$_2$ concentrations over the study domain during the study period are shown in Fig. 5(a). The spatial variability of near-surface NO$_2$ correlates with Germany's population density, and the main hotspots correspond to Germany's major metropolitan areas (Figure A3). The study domain's main hotspot is western Germany (North Rhine-Westphalia; a federal state of Germany), Germany's industrial heartland. The number of days (%) that exceeded the 2021 WHO NO$_2$ limit (24-hr mean > 25 $\mu$g m$^{-3}$) over major metropolitan areas in Germany was more than

50%, with western Germany exceeding the WHO NO$_2$ limit on more than 80% of the days during the study period (Fig. 7). Around 36% of people live in locations where more than 25% of days exceed the WHO NO$_2$ limit during the study period (Fig. 8). The GBT-simulated near-surface O$_3$ showed distinct spatial variability compared to NO$_2$, with high O$_3$ concentrations over southern Germany and low O$_3$ concentrations over northern Germany (Fig. 6). This could be due to the fact that O$_3$ is a secondary pollutant that is primarily driven by photochemical reactions influenced by meteorology; DUV and temperature

values, which were the most influencing factors for photochemical reactions and accordingly the most important features fitted in the O$_3$ GBT model, were higher in southern Germany than northern Germany (Figure A4). During the study period, more





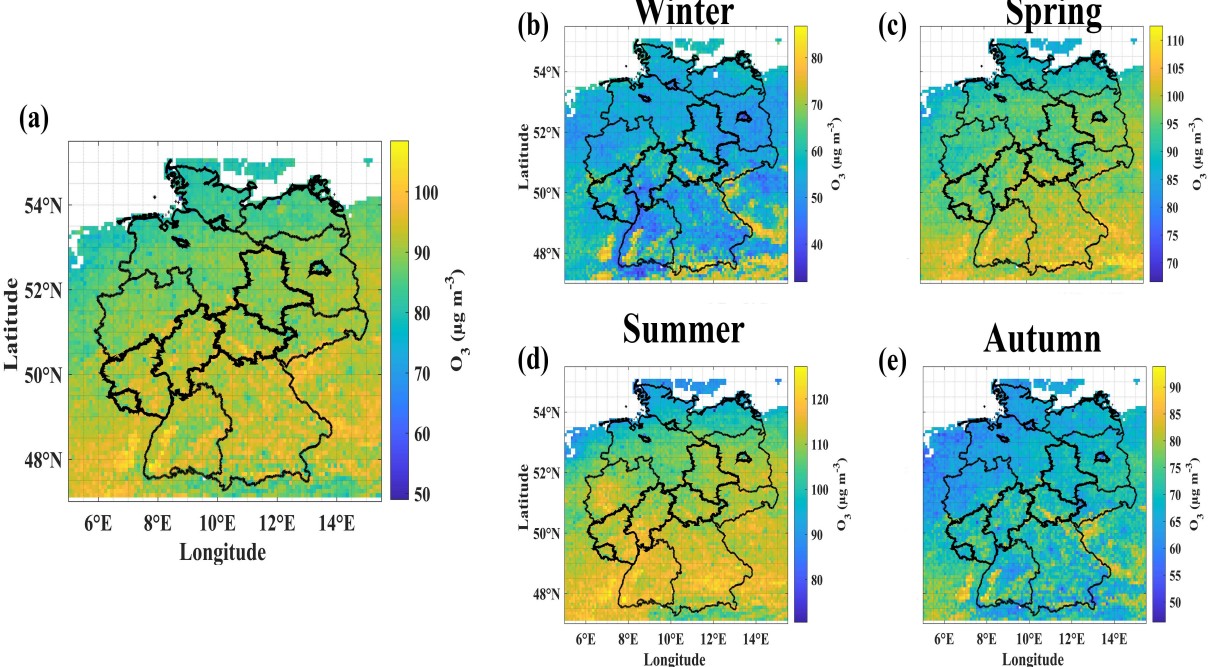

**Figure 6.** (a) Averaged GBT-simulated near-surface $O_3$ concentrations over the study domain during the study period. (b-e) Averaged GBT-simulated near-surface $O_3$ concentrations for each season during the study period.

than 50% of days in southern Germany exceeded the 2021 WHO $O_3$ limit (maximum 8-hr mean > 100 $\mu$g m$^{-3}$). Nearly 90% of people live in locations where more than 25% of days exceed the WHO $O_3$ limit (Fig. 8). Another interesting fact is that southern metropolitan areas and high NO$_X$ regions have less days that exceeded the WHO $O_3$ limit than southern rural regions

(Fig. 7). It is a well-known fact that rural regions have higher ozone levels than urban regions (Malashock et al., 2022). It could be because NO is a significant $O_3$ scavenger in higher NO$_X$ (NO$_2$ is a proxy for NO$_X$) regions or due to being in a NO$_X$ saturated regime. Furthermore, it is due to the fact that rural regions being the downwind locations of emission plume and are the primary source of biogenic VOC emissions (Zong et al., 2018).

The GBT-simulated near-surface $NO_2$ showed seasonal variations, as expected, with higher values in the winter season

(Fig. 5). This is because of high-residential heating demand and favorable meteorology (e.g., a low boundary layer height) for pollutant accumulation and less $NO_2$ photolysis due to low solar radiation in the winter. The near-surface $NO_2$ hotspots were the same in all seasons, as seen in the overall study period average. In contrast, near-surface $O_3$ showed strong seasonal variations, with high values in the spring and summer due to high solar radiation (Fig. 6). It is worth noting that, as seen in the overall study period average, $O_3$ values in southern Germany were significantly higher in spring and summer than in

northern Germany. Because near-surface $O_3$ is mainly driven by meteorology (DUV and temperature, which drive photochemical reactions and precursor emissions), the spatial and temporal variability is attributed to changes in meteorology. We also compared the spatial variability of GBT-simulated near-surface $NO_2$ and $O_3$ to the CAMS reanalysis product for the period



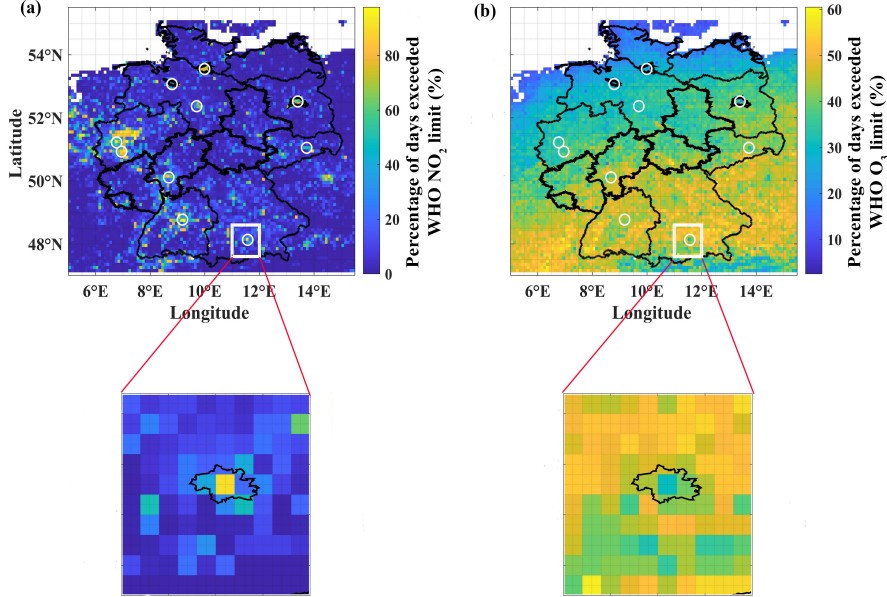

**Figure 7.** Number of days (%) that exceeded the WHO 24-hr mean $NO_2$ (a) and maximum 8-hr mean $O_3$ (b) limits over the study domain during the study period based on GBT-model simulations. White circles represent major metropolitan areas. The metropolitan area of Munich and its surroundings (rectangular box) are enlarged to illustrate the urban vs. rural gradient. The administrative boundaries of Munich are marked in black in the inset panel.

between 2019-07-17 and 2020-01-31 (Figure A5 and A6). The spatial variability of GBT-simulated near-surface $NO_2$ and $O_3$ agreed well with CAMS model. This implies that the ML model can supplement or replace the computationally expensive
chemical transport models.

### 3.4 Influence of COVID-19 lockdown restrictions on near-surface $NO_2$ and $O_3$ changes

Due to the COVID-19 out-break, many nations, including Germany, announced a lockdown in the spring of 2020. During that time period, various anthropogenic emission activities were restricted, affecting particularly traffic-related emissions. To estimate the influence the lockdown restrictions on air pollutant concentration changes, we compared the GBT-simulated 2020
lockdown concentration with the same period in 2019. The 2020 lockdown period measurements were not used for GBT model training in this comparison. This can also be regarded as the critical performance evaluation of the GBT model.

When comparing different time periods, it is crucial to account for meteorological effects when estimating the impact of anthropogenic emission reductions (i.e., lockdown effects) on changes in air pollutant concentrations. Therefore, as described in the method section, we used GC simulations to exclude the meteorology contribution from GBT-simulated concentrations.
After disentangling the meteorology contribution, it is noticeable that high near-surface $NO_2$ levels decreased primarily over





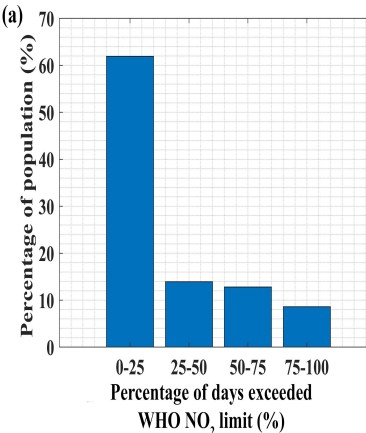
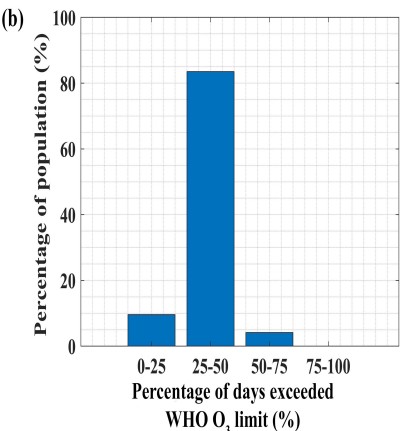

**Figure 8.** The population distribution in terms of the number of days (%) that exceeded the WHO 24-hr mean $NO_2$ (a) and maximum 8-hr mean $O_3$ (b) limits over the study domain during the study period based on GBT-model simulations.

the previously observed hotspots (Fig. 9). The near-surface $O_3$ increased over western Germany while decreasing elsewhere, particularly over low $NO_X$ regions. We already observed that western Germany was a $NO_X$ hotspot, possibly a $NO_X$ saturated regime, so a reduction in $NO_X$ increases ozone. Also, we could see that changes in near-surface $O_3$ were either negligible or slightly increased over metropolitan areas. The meteorology-accounted for mean lockdown near-surface $NO_2$ decreased by about 23 ($\pm5.3$)%, while meteorology-accounted for mean lockdown near-surface $O_3$ increased by 1 ($\pm4.6$)%, over ten major metropolitan areas (Berlin, Bremen, Cologne, Dresden, Düsseldorf, Frankfurt, Hamburg, Hanover, Munich, and Stuttgart), compared to 2019. It increased by about 9% in the Cologne and Düsseldorf metropolitan areas (located in western Germany) and slightly increased or decreased (between -3 and +2%) in other metropolitan areas, compared to 2019. This finding is consistent with other studies that found a decrease in meteorology-accounted for lockdown near-surface $NO_2$ and the small increase in lockdown near-surface $O_3$ over German metropolitan areas compared to 2019 using in-situ measurements (Balamurugan et al., 2021, 2022b). We also evaluated our GBT model's ability to represent different emission scenarios by comparing weekends and weekdays; typically, anthropogenic $NO_X$ emissions on weekends are lower than on weekdays due to reduced vehicle transportation. Our GBT model was also able to distinguish between the weekend and weekday emission scenarios; weekend near-surface $NO_2$ was lower than weekday near-surface $NO_2$, and, as expected, there were no or only slight changes in weekend near-surface $O_3$ compared to weekdays, with slight increases particularly over metropolitan areas (Figure A7).

### 3.5 Transferability of our GBT model

Our study domain also covered parts of other European countries. However, we trained our GBT model using data from German measurement stations only. Therefore, comparing our trained GBT model simulations with measurements in other countries demonstrates how well our GBT model models near-surface $NO_2$ and $O_3$ concentrations in neighboring parts of the world; similar to the location-leave-out testing strategy. We chose five major cities (Salzburg, Prague, Strasbourg, Liège,





Absolute changes in GBT-simulated near-surface NO$_2$ and O$_3$ concentrations in 2020 lockdown period compared to 2019

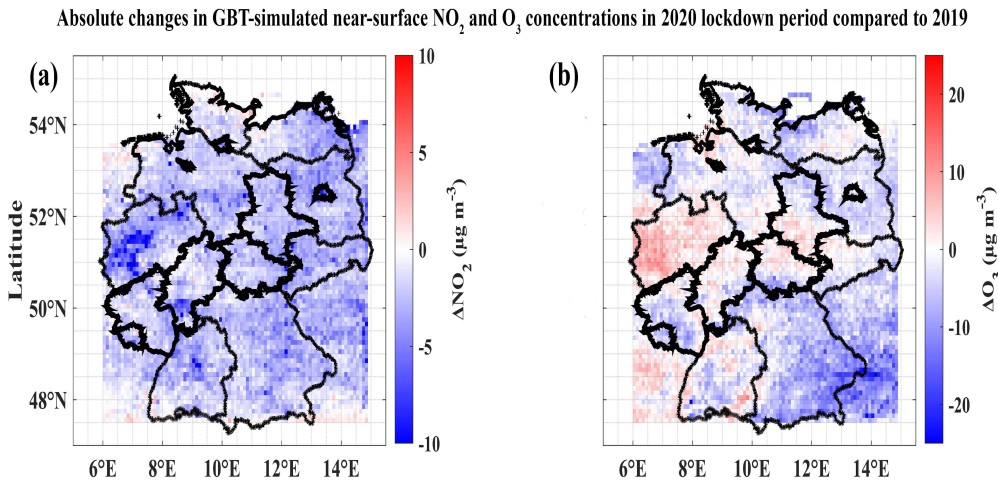

**Figure 9.** Absolute changes in GBT-simulated near-surface NO$_2$ and O$_3$ concentrations in 2020 lockdown period compared to the same period in 2019 after accounting for meteorology.

and Groningen) in different European countries covered by our study domain and compared their measured NO$_2$ and O$_3$ concentrations with GBT modeled NO$_2$ and O$_3$ concentrations (Fig. 10 & Table A5).

Our trained NO$_2$ GBT model based on German measurement stations explained 32-64% (R$^2$ ranges between 0.32 and 0.64, and RMSE ranges between 9.76 and 13 $\mu$g m$^{-3}$) of near-surface NO$_2$ measured in five metropolitan areas located outside of

Germany, while O$_3$ GBT model simulations agreed well with observations (R$^2$ ranges between 0.87 and 0.94, and RMSE ranges between 9.55 and 14.32 $\mu$g m$^{-3}$). Since near-surface O$_3$ is mainly driven by meteorology, the O$_3$ GBT model trained using German measurement stations explains a large portion of near-surface O$_3$ in other locations. The worse agreement between NO$_2$ GBT model predictions and NO$_2$ observations in other European countries suggests that information is lacking in the NO$_2$ GBT model for better representation of other locations, similar to location-leave-out 5-fold CV, which also showed

low agreement for the NO$_2$ GBT model when modeling new locations (Table 2). Differences in vehicle fleet composition and emission standards across different countries/locations would have an impact on our NO$_2$ GBT model predictions when applied to other countries/locations. In future work, other features/proxies besides road density could be considered to represent traffic emission.

## 4 Conclusion

This study simulated near-surface NO$_2$ and O$_3$ concentrations using an ML model over Germany at 0.1 degree resolution and daily intervals. The ML model was used to link satellite column measurements (proxies for near-surface air pollutants), meteorology and proxies of emission source information to near-surface NO$_2$ and O$_3$ concentrations. The ML models are



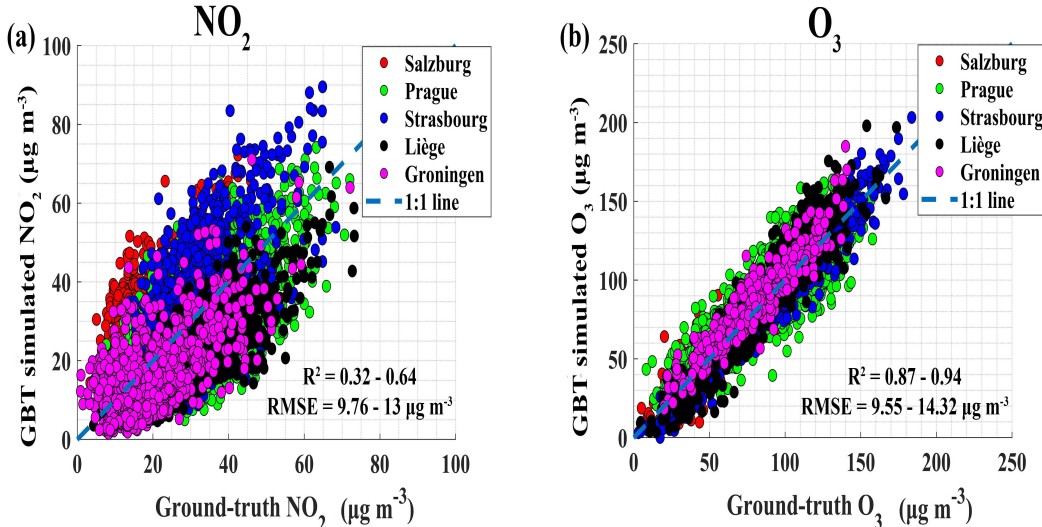

**Figure 10.** Comparison between ground-truth and GBT-simulated near-surface $NO_2$ (a) and $O_3$ (b) for five different European metropolitan areas.

extremely effective at learning the complex non-linear relationships between variables. Therefore, in this study, we explored the capabilities of ML models in the spatio-temporal prediction of air pollutants. In addition, we investigated three aspects of

the ML model: 1. how well our ML model performs compared to the chemical transport model, 2. how well our ML model can be used to assess the effectiveness of mitigation initiatives; and 3. how well our ML model can be transferred to locations where measurements are unavailable.

Four different testing strategies were performed to evaluate the ML model's spatio-temporal prediction: 1. Random split of data (70% for training and 30% for testing), 2. Random 5-fold CV, 3. Time-leave-out 5-fold CV, and 4. Location-leave-out

5-fold CV. The gradient boosted tree (GBT) model trained for $NO_2$ explained about 68-88% of observed $NO_2$ concentrations in Germany, with RMSE of 4.77-8.67 $\mu g$ m$^{-3}$, whereas the GBT model trained for $O_3$ performed even better, with an $R^2$ of 0.74-0.92 and RMSE of 8.53-13.2 $\mu g$ m$^{-3}$. The evaluation metrics of the GBT model for different testing strategies differed significantly. The location-leave-out 5-fold CV testing strategy showed poor agreement for the $NO_2$ GBT model, whereas the time-leave-out 5-fold CV testing strategy showed poor agreement for the $O_3$ GBT model. This points out the importance

of performing different testing strategies to interpret the true capability of the ML model. The road $NO_X$ emission source proxy (road density) and TROPOMI tropospheric column $NO_2$ were the most important features in the fitted $NO_2$ GBT model. However, for $O_3$, the most important features were downward UV radiation at the surface and temperature. The multi-layer perceptron (MLP) model trained for both $NO_2$ and $O_3$ performed worse than the GBT model.

We also showed that our $NO_2$ GBT model outperforms the CAMS model, while slightly under-performing for near-surface

$O_3$. The CAMS model reanalysis data set uses real-time observations with an ensemble of eleven air-quality models through data assimilation techniques, which are expected to be more computationally expensive than our GBT model. Therefore, the





spatio-temporal variability of near-surface $NO_2$ and $O_3$ concentrations and human exposure at a locations where no measurements are available can be studied with lower computational effort when using our GBT model. Near-surface $NO_2$ hotspots were found over German metropolitan areas, particularly western Germany. The near-surface $NO_2$ hotspots locations did not change with the seasons but had high values in the winter. However, near-surface $O_3$ showed high seasonal variability, with high values in the spring and summer and no definite hotspots. Overall, southern Germany experiences higher ozone levels than northern Germany due to higher downward UV radiation and temperatures in southern Germany compared to northern Germany. Even though metropolitan areas were the $NO_2$ hotspots, rural regions, particularly in southern Germany, had higher $O_3$ concentrations than metropolitan areas. It is because rural areas are dominated by meteorology-driven biogenic VOC emissions and are generally situated downwind of the emission plume. About 36% of people live in locations where WHO $NO_2$ limit exceeds more than 25% of days during the study period. Meanwhile, 90% of the people lives in areas where the WHO $O_3$ limit is exceeded for more than 25% of days.

Our study also demonstrated the GBT model's capability to assess the efficacy of mitigation strategies. For example, our GBT model reproduced the observations that, during the 2020 COVID-19 lockdown period, meteorology-accounted for near-surface $NO_2$ was significantly reduced, while meteorology-accounted for near-surface $O_3$ was slightly increased or decreased over metropolitan and industrial areas over Germany, compared to 2019. These findings agreed with those of other studies that used in-situ measurements.

Our GBT ML model's transferability is assessed by comparing simulations from our GBT model trained with measurements in Germany to measurements in other European countries. Our $NO_2$ GBT model showed moderate agreement with observations from other countries ($R^2$ ranges between 0.32 and 0.64, and RMSE ranges between 9.76 and 13 $\mu$g m$^{-3}$), implying a lack of information in the GBT model when modeling near-surface $NO_2$ over other countries, which may have different vehicle fleet composition and emissions standards. However, our $O_3$ GBT model performed well ($R^2$ ranges between 0.87 and 0.94, and RMSE ranges between 9.55 and 14.32 $\mu$g m$^{-3}$), indicating that our $O_3$ GBT model can be used to model the $O_3$ concentrations in other countries, at least in neighboring European countries.

*Code and data availability.* The various data sets and code used to conduct this study will be made available on GitHub following publication.





## Appendix A

**Table A1.** Different type of stations (%) considered in this study (based on locations specified by the European Environment Agency).

|  | Traffic | Industrial | Background |
|---|---|---|---|
| **Near-surface NO$_2$** | 37.1% | 5.3% | 57.6% |
| **Near-surface O$_3$** | 2.7% | 5.8% | 91.4% |

**Table A2.** The hyperparameters of the GBT model for each pollutant used in the study.

| Hyper paramertes | NO$_2$ model | O$_3$ model |
|---|---|---|
| **Max_depth** | 10 | 10 |
| **Learning_rate** | 0.3 | 0.3 |
| **reg_lambda** | 12 | 4 |
| **reg_alpha** | 18 | 26 |
| **gamma** | 20 | 8 |
| **min_child_weight** | 16 | 8 |
| **n_estimators** | 2500 | 2500 |

**Table A3.** The hyperparameters of the MLP model for each pollutant used in the study.

| Hyper paramertes | NO$_2$ model | O$_3$ model |
|---|---|---|
| **Hiddern_layers** (neurons in each layer) | 3 (200,100,50) | 4 (350,150,75,37) |
| **activation** | tanh | tanh |
| **alpha** | 0.04 | 0.1 |
| **learning rate** | adaptive | adaptive |
| **solver** | sgd | lbfgs |
| **Max_iter** | 2000 | 1500 |



**Table A4.** Evaluation metrics of our MLP model in different testing strategies.

|  |  | Random (70%/30%) | Random (5-fold) | Time-leave-out (5-fold) | Location-leave-out (5-fold) |
|---|---|---|---|---|---|
| **NO$_2$** | **R$^2$** | 0.79 | 0.82±0.006 | 0.54±0.29 | 0.46±0.25 |
| **MLP model** | **RMSE ($\mu$g m$^{-3}$)** | 4.77 | 5.9±0.11 | 8.6±1.76 | 13.2±1.07 |
| **O$_3$** | **R$^2$** | 0.83 | 0.9±0.001 | 0.42±0.37 | 0.71±0.13 |
| **MLP model** | **RMSE ($\mu$g m$^{-3}$)** | 12.15 | 9.6±0.027 | 20.1±7.3 | 14.9±3.2 |

**Table A5.** Metropolitan areas in other European cities considered for the evaluation of GBT model. The evaluation metrics (comparison between GBT simulations and in-situ measurements) for NO$_2$ and O$_3$ shown in last two columns for each city.

| Metropolitan area (country) | Coordinates | R$^2$ and RMSE ($\mu$g m$^{-3}$) for NO$_2$ | R$^2$ and RMSE ($\mu$g m$^{-3}$) for O$_3$ |
|---|---|---|---|
| **Salzburg (Austria)** | 47.80° N, 13.05° E | 0.32 and 12.52 | 0.87 and 12.43 |
| **Prague (Czech Republic)** | 50.07° N, 14.43° E | 0.43 and 10.05 | 0.79 and 14.32 |
| **Strasbourg (France)** | 48.57° N, 7.75° E | 0.47 and 13 | 0.94 and 9.55 |
| **Liège (Belgium)** | 50.63° N, 5.56° E | 0.64 and 11.9 | 0.88 and 12.04 |
| **Groningen (Netherlands)** | 53.21° N, 6.56° E | 0.34 and 9.76 | 0.87 and 11.33 |

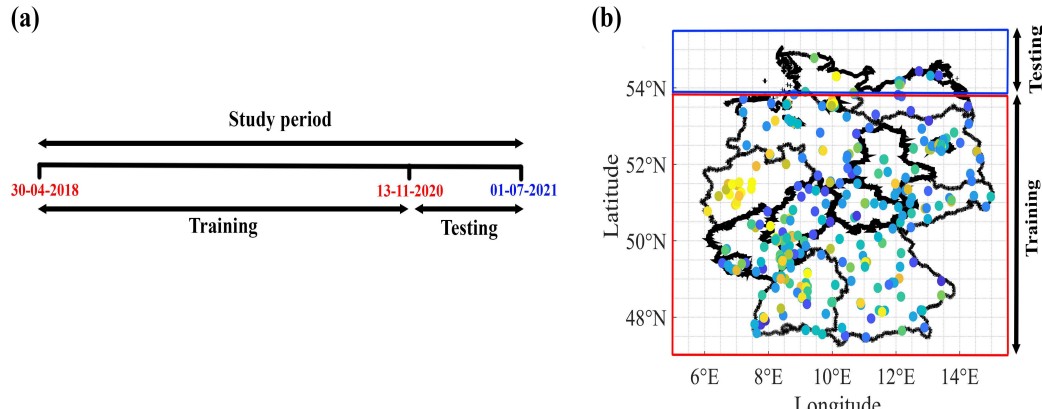

**Figure A1.** A first one-fold step in 5-fold CV is illustrated for time-leave-out (a) and location-leave-out (b) testing strategies. In time-leave-out 5-fold CV, the data was divided into 5 parts based on time period (date-wise), with four parts used for training and one part tested. This process is repeated until each part (a total of 5) has been tested. Similarly, in location-leave-out 5-fold CV, the data was divided into 5 parts based on location (latitude), with four parts used for training and one part tested. This process is repeated until each part (a total of 5) has been tested.



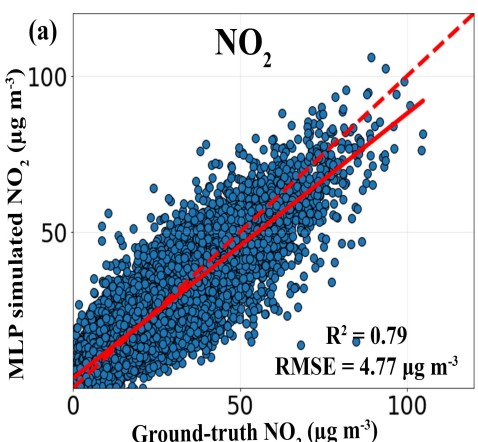

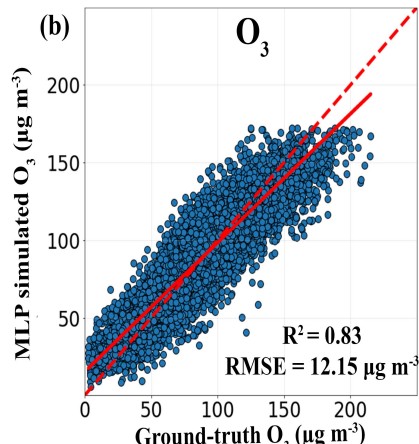

**Figure A2.** Comparison between ground-truth and MLP-simulated near-surface $NO_2$ (a) and $O_3$ (b). The dotted line represents a 1:1 line, while the solid line represents a linear fit.

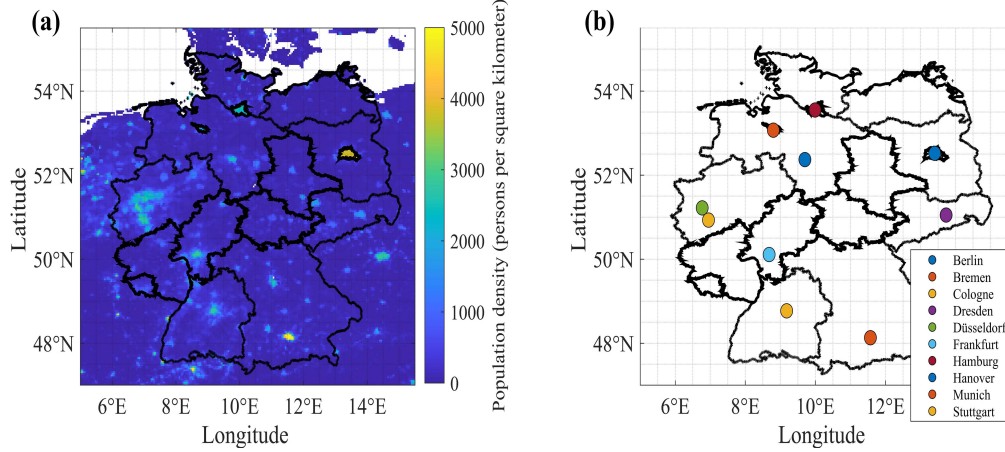

**Figure A3.** Population density for the year 2020 (a) and the locations of major German metropolitan areas (b).



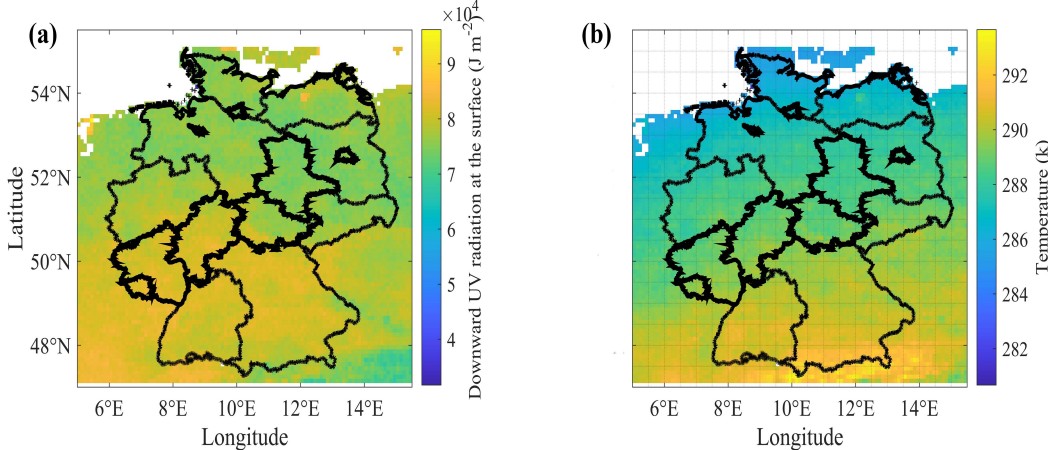

**Figure A4.** Averaged "Downward UV radiation at the surface" (a) and "Temperature" (b) over the study domain during the study period.

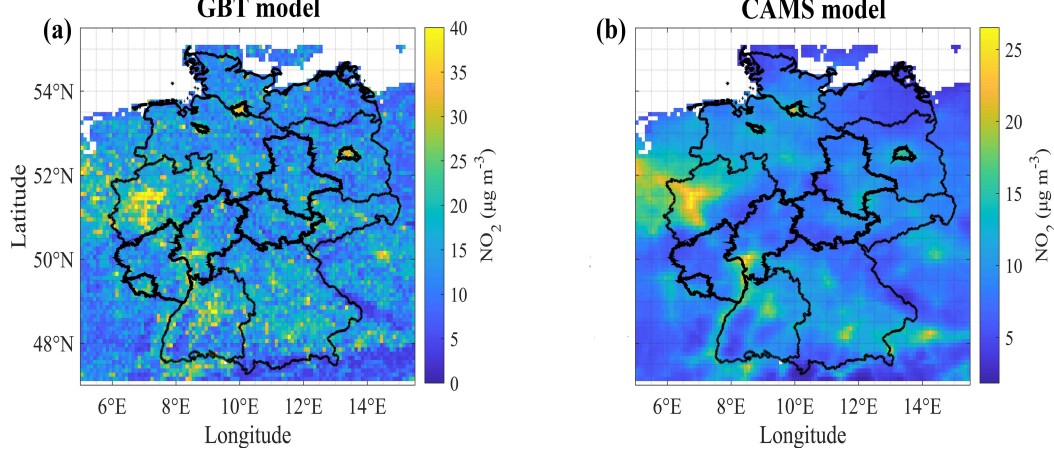

**Figure A5.** Averaged GBT-simulated near-surface $NO_2$ concentrations (a) and CAMS reanalysis near-surface $NO_2$ concentrations (b) over the study domain for the period between 2019-07-17 and 2020-31-01.



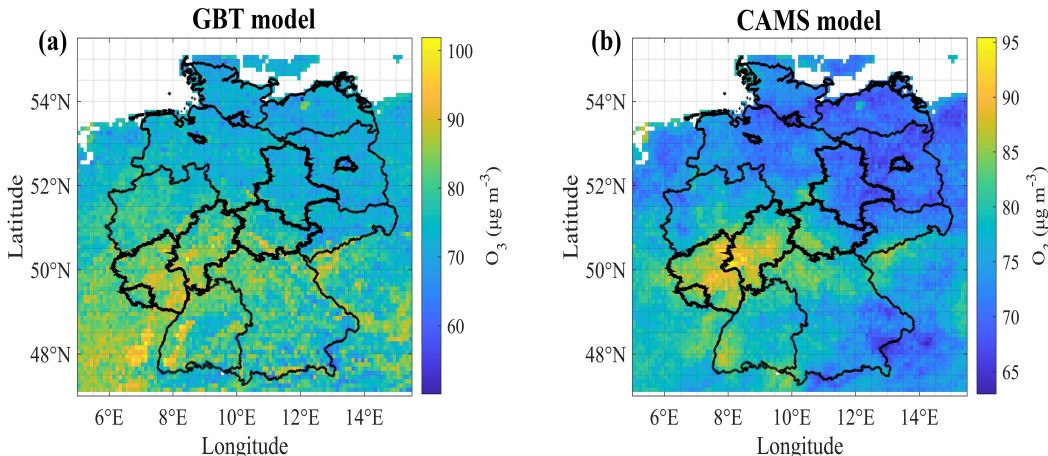

**Figure A6.** Averaged GBT-simulated near-surface $O_3$ concentrations (a) and CAMS reanalysis near-surface $O_3$ concentrations (b) over the study domain for the period between 2019-07-17 and 2020-31-01.

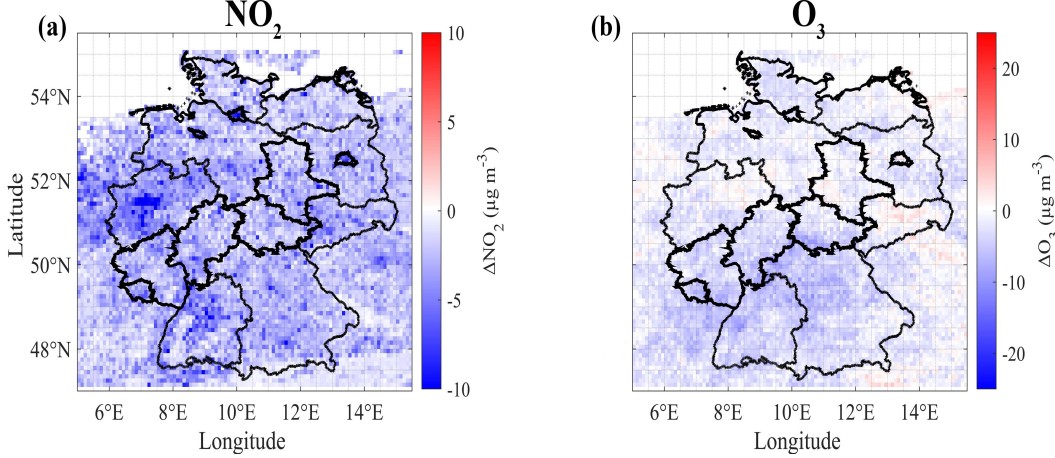

**Figure A7.** The difference in GBT-simulated near-surface $NO_2$ (a) and $O_3$ (b) concentrations between weekend and weekday during the study period.



*Author contributions.* VB, JC and FNK conceived the study and designed the concept. VB obtained all of the data, performed the modelling work and analysed the results. VB and AW developed the methodology. JC and FNK acquired the funding and supervised the work. VB

370   wrote the manuscript. JC, AW and FNK reviewed and edited the manuscript

*Competing interests.* The authors declare that they have no conflict of interest.

*Acknowledgements.* This research has been funded by the Institute for Advanced Study, Technical University of Munich (grant no. 291763).

The authors thank the European Environment Agency, the Copernicus Services, the GES DISC data archive and the United States Geological Survey for providing free access to the various data sets used in this study.



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
