# Peer review of "Spatio-temporal modeling of air pollutant concentrations in Germany using machine learning"

_EGUsphere, 2023_

## Author Comment (AC1)

Dear Reviewer,

We appreciate your comments and suggestions, which have helped us improve our manuscript further. We have made the necessary changes to the manuscript, which can be found in the attached file (Track Changes). The following is a response to your comments and suggestions. Corresponding changes in the revised manuscript are also made available below, if applicable, at the appropriate places.

Sincerely,

On behalf of all co-authors,

Vigneshkumar Balamurugan

─────────────────────────────────────────────────────────

**Response to Reviewer-1:**

**The authors explored the gradient boosted tree approach for spatial-temporal modelling of NO2 and O3 and applied it to the case in Germany. There are some issues to address in the revised version:**

Thank you so much for reading and reviewing our manuscript! We carefully reviewed and considered your comments/suggestions, and made improvements in the revised manuscript.

**Validations:**

**Table 1 lists the types of datasets used in this study. May you clarify which dataset was used for the ground-truth data?**

In the revised manuscript (Table 1), we now included the purpose of the data.

Table 1. Data sets and related information used in this study.

| Data source | Data (purpose) | Temporal resolution | Spatial resolution |
|---|---|---|---|
| Governmental in situ measurements | Near-surface $NO_2$ and $O_3$ (Ground-truth data) | 1 hr | - |
| TROPOMI satellite measurements | Tropospheric column $NO_2$, total column $O_3$ and total column HCHO (Input features) | Daily | 7 km*3.5 km (5.5 km*3.5 km, after 6 August 2019) |
| ERA5 (ECMWF reanalysis) | Temperature, relative humidity, wind speed, wind direction, downwind UV solar radiation at surface, boundary layer height, surface pressure and temperature of air at 2m above the surface (Input features) | 1 hr | 0.25*0.25-degree |
| U.S. Geological Survey | Surface elevation (Input features) | - | 1*1-km |
| GRIP global roads database | Road density (Input features) | - | 8*8-km |
| CAMS European air quality forecasts | Near-surface $NO_2$ and $O_3$ (for validation) | 1 hr | 0.1*0.1-degree |

| GEOS-Chem chemical transport model | Near-surface NO$_2$ and O$_3$ (for disentangling meteorology impacts) | 1 hr | 0.5*0.625-degree |
|---|---|---|---|

**Figures 5-6 show the spatial distribution of the averaged NO2 and O3 during the study period. Is the study period between 2019-07-17 and 2020-01-31? May you specify which months were used for Summer, Spring, Autumn, and Winter?**

In figure 5 (and 6), the averaged NO$_2$ (and O$_3$) concentrations are between 2018-04-30 and 2021-07-01. We have updated the figure captions in both Figure 5 and Figure 6 to include the study period as well as the specific months used to calculate the seasonal averages.

Figure 5. (a) Averaged GBT-simulated daily near-surface NO$_2$ concentrations over the study domain for the study period between 2018-04-30 and 2021-07-01. (b-e) Averaged GBT-simulated daily near-surface NO$_2$ concentrations for each season during the study period. Winter: December, January and February. Spring: March, April and May. Summer: June, July and August. Autumn: September, October and November.

Figure 6. (a) Averaged GBT-simulated daily near-surface O$_3$ concentrations over the study domain for the study period between 2018-04-30 and 2021-07-01. (b-e) Averaged GBT-simulated daily near-surface O$_3$ concentrations for each season during the study period. Winter: December, January and February. Spring: March, April and May. Summer: June, July and August. Autumn: September, October and November.

**The data sets were pre-processed in daily scale. Could you please generate a spatial map illustrating the average daily concentrations of NO2 and O3 during Summer and Winter, instead of considering the seasonal averages? Furthermore, may you compare these results with reanalysis from CAMS?**

For Figures 5 and 6, the seasonally average $NO_2$ (and $O_3$) values were not simulated. The Machine learning model was used to simulate daily $NO_2$ and $O_3$ concentrations spatial map, and daily maps were averaged for each season, as shown in Figure 5 (and 6). We also modified the figure 5 and 6 captions to  make it clearer to the reader. We hope this clarifies your comment.

Figure 5. (a) Averaged GBT-simulated daily near-surface $NO_2$ concentrations over the study domain for the study period between 2018-04-30 and 2021-07-01. (b-e) Averaged GBT-simulated daily near-surface $NO_2$ concentrations for each season during the study period. Winter: December, January and February. Spring: March, April and May. Summer: June, July and August. Autumn: September, October and November.

Figure 6. (a) Averaged GBT-simulated daily near-surface $O_3$ concentrations over the study domain for the study period between 2018-04-30 and 2021-07-01. (b-e) Averaged GBT-simulated daily near-surface $O_3$ concentrations for each season during the study period. Winter: December, January and February. Spring: March, April and May. Summer: June, July and August. Autumn: September, October and November.

CAMS European air quality forecasts are only available for three years in the rolling archive. Therefore, we only compare the CAMS product for the period between 2019-07-17 and 2020-31-01 (Figure A5 and A6).

**Line 131, "we also included "Near-surface NO2" modeled from NO2 ML model as a feature variable in the O3 ML model." However, in Figure 3 (d), the Near-surface NO2" modeled from NO2 ML model is not listed. I guess the Near-surface NO2" modeled from NO2 ML model will be top one affecting the O3 predive results. Is this case? Maybe you can use the ML model to get the direct relationship between O3 and Near-surface NO2" modeled from NO2 ML model.**

Yes. We agree with the reviewer that ML modeled near-surface $NO_2$ is one of the most important factors influencing $O_3$ predictive results. Based on our results, it is the sixth most important feature. In figure 3(d), "ML modeled near-surface $NO_2$" is given as "in-situ $NO_2$". This is changed in the revised manuscript (Figure 3).

When using machine learning models, the direct relationship between variables, such as $NO_2$ and $O_3$, cannot be obtained as deterministic equations. Instead, one can analyze the feature importance or variable importance provided by the model.

[Figure]

Figure 3. Comparison between ground-truth and GBT-simulated near-surface $NO_2$ (a) and $O_3$ (b). Feature importance (top 10) calculated based on SHAP (SHapley Additive exPlanations)

values for $NO_2$ (c) and $O_3$ (d) GBT model. RD: Road Density, BLH: Boundary Layer Height, E: Surface Elevation, T: Temperature, DOW: Day of the week, RH: Relative Humidity, T2m: Temperature at 2 meter height, DUV: Downwind UV radiation, WS: Wind speed, WD: Wind Direction.

**Line 243, "After the discussed model evaluation, we trained the GBT model using 100% of the data and modeled the near-surface NO2 and O3 concentrations over the study domain at 0.1 degree resolution and daily", It is not clear here. Are you re-train the model? How do you validate your model?**

Yes. We trained the model using 100% of data after performing different ML model evaluations, which is a common practice in machine learning to leverage all available information and avoid losing any valuable data. After using 100% of the data for training, the model can only be evaluated with ground-truth data beyond the study period. However, in our ML model evaluations, we followed different validation approaches that involved more than just training and evaluating a single model. For example, we employed evaluation strategies such as the five-fold random/time/location-leave-out method. These methods enabled us to train five ML models by systematically leaving out different subsets of the whole dataset during each fold validations. Therefore, we believe our ML model would perform similarly on future data, as the models' performance on unseen data yielded robust estimates of their generalization ability during the different evaluation strategies.
* * *
**Response to Reviewer-2:**

**General comments**

**The authors develop a machine learning framework for modeling NO2 and O3 concentrations in Germany, and based on that, they analyze human exposure to the two air pollutants and the effects of COVID quarantine. The authors also discuss the transferability of their model.**

**The manuscript is well organized and in particular the methodology is thoroughly described. However, before it can be published, I believe the authors should address the comments below.**

Thank you so much for taking the time to read and review our manuscript! We carefully reviewed and considered your comments/suggestions, and as a result, we improved the manuscript.

**Specific comments**

**Line 129: Does the "season" (season of the year) information in the ML model have only 4 values? In my opinion, "day of the year" would be a more ideal feature to help the model learn the daily variability of air pollutants. The author should try or clarify this.**

Thank you for your suggestion! We have evaluated the ML model using both "Day of the Year" and "season of the year" as features in all our evaluation strategies. We noted that there is a slightly worse performance in both the $NO_2$ and $O_3$ GBT model (Table R1 and R2), when using "Day of the year" as a feature instead of "season of the year". Therefore, we decided to use "season of the year" instead of "Day of the Year" in our study.

Table R1. Evaluation metrics of our GBT model in different testing strategies (using "Season of the Year" as a feature).

|  |  | Random (1-fold) | Random (5-fold) | Time-leave-out (5-fold) | Location-leave-out (5-fold) |
|---|---|---|---|---|---|
| $NO_2$ GBT model | $R^2$ | 0.88 | 0.89±0.002 | 0.74±0.07 | 0.68±0.12 |
|  | RMSE (µg $m^{-3}$) | 4.77 | 4.65±0.034 | 6.77±0.7 | 8.67±1 |
| $O_3$ GBT model | $R^2$ | 0.92 | 0.92±0.001 | 0.74±0.09 | 0.8±0.06 |
|  | RMSE (µg $m^{-3}$) | 8.53 | 9.36±0.068 | 13.2±1.01 | 12.45±1.26 |

Table R2. Evaluation metrics of our GBT model in different testing strategies (using "Day of the Year" as a feature).

|  |  | Random (1-fold) | Random (5-fold) | Time-leave-out (5-fold) | Location-leave-out (5-fold) |
|---|---|---|---|---|---|
| $NO_2$ GBT model | $R^2$ | 0.88 | 0.89±0.002 | 0.74±0.061 | 0.68±0.14 |
|  | RMSE (µg $m^{-3}$) | 4.76 | 4.67±0.05 | 6.76±0.68 | 8.74±1.3 |
| $O_3$ GBT model | $R^2$ | 0.91 | 0.90±0.001 | 0.72±0.09 | 0.78±0.06 |
|  | RMSE (µg $m^{-3}$) | 8.60 | 9.82±0.054 | 13.6±1.16 | 12.96±1.21 |

**Line 131: Given the coupled nature of NO2 and ozone, I would suggest the authors try to include O3 as a feature in the NO2 ML model, like why they did the same way for O3 model, or please clarify why they didn't do so.**

If we include the ML-modeled $O_3$ in the $NO_2$ ML model iteratively, we believe the ML model may suffer from overfitting. For example, $O_3$ could become an important feature as it already contains information about $NO_2$ (it is important to note that the ML-modeled $NO_2$ is the sixth most important feature). Additionally, the errors from both the $NO_2$ and $O_3$ ML models in the first iteration would propagate and potentially amplify the errors.

**Line 148: 24h-mean of ERA-5 data makes sense for NO2 model, but I would suggest the authors to test daytime-mean or daily-max for O3 model, as ozone is calculated as MDA8. This is especially the case for daily-max 2m temperature, which has been shown to be well correlated with MDA8 ozone.**

Thanks for the suggestion! Before deciding on the 24-hr mean of meteorology as a feature for the $O_3$ GBT model, we also conducted a test on the maximum $O_3$-time (10 - 6 local time), when maximum 8-hr $O_3$ concentration occurs (Figure R1). When we used maximum $O_3$-time mean as a feature, we noted a similar performance, compared to 24-hr mean as a feature (Table R3 and R4). Therefore, we chose a 24-hour mean for both the $NO_2$ and $O_3$ models.

[Figure]

Figure R1. The diurnal mean $O_3$ averaged between 2010 and 2019.

Table R3. Evaluation metrics of our $O_3$ GBT model in different testing strategies (using "24-hr mean of meteorology" as a feature).

| | | Random (1-fold) | Random (5-fold) | Time-leave-out (5-fold) | Location-leave-out (5-fold) |
|---|---|---|---|---|---|
| $O_3$ GBT model | $R^2$ | 0.92 | 0.92±0.001 | 0.74±0.09 | 0.8±0.06 |
| | RMSE (µg m$^{-3}$) | 8.53 | 9.36±0.068 | 13.2±1.01 | 12.45±1.26 |

Table R4. Evaluation metrics of our $O_3$ GBT model in different testing strategies (using "maximum $O_3$-time of meteorology" as a feature).

| | | Random (1-fold) | Random (5-fold) | Time-leave-out (5-fold) | Location-leave-out (5-fold) |
|---|---|---|---|---|---|
| $O_3$ GBT model | $R^2$ | 0.91 | 0.92±0.001 | 0.75±0.09 | 0.79±0.07 |
| | RMSE (µg m$^{-3}$) | 8.6 | 8.8±0.054 | 13.16±1.16 | 12.65±1.47 |

**Line 160: Authors should give the exact size of data samples (both training and testing set), as text or labelled on the figure.**

In the revised manuscript, we added the training and test sample size in the corresponding locations.

| Line 196-197 | The trained GBT model with 70% of the data (78433) for $NO_2$ reproduced the observed $NO_2$ concentration well in the test case (33615), with an $R^2$ of 0.88 and RMSE of 4.77 µg m$^{-3}$. |
|---|---|

| Line 215-216 | The GBT model trained with 70% of the data (65705) for $O_3$ also well reproduced the observed $O_3$ concentrations in the test case (28160), with an $R^2$ of 0.92 and RMSE of 8.53 µg m$^{-3}$. |
|---|---|

**Line 205: It is interesting to see that road density is the most important feature, given that it has constant values which don't show temporal variations. Can the authors explain this further?**

We agree with the reviewer that road density doesn't show temporal variation for a particular location. However, in our study, we developed a ML model for the whole Germany domain, in which spatial variation in road density explains the majority of the near-surface $NO_2$ variation. Therefore, road density is the most important feature in our ML model.

**Line 229 (and also line 153): The fact that MLP is worse than GBT can be interesting or maybe controversial here, as people now tend to believe that deep learning techniques should outperform light-weight algorithms such as GBT. The authors should explain more about this, as it is an important and perhaps new finding. Personally, I can think of a few questions below that might help clarify this.**

- **What is tabular/structured data and what is non-tabular/structured data? Is the data we use for air pollutants prediction usually of the former type?**

  In this study, we prepared the data as structured data format. Tabular/Structured Data and Non-Tabular/Unstructured Data are the terms used to categorize different types of data based on their format. Tabular/structured data refers to data that is organized in a tabular format, similar to a table or spreadsheet. Most ML models, such as decision trees, SVR, and feedforward neural networks, take this type of input.

  Non-tabular/unstructured data refers to data that does not have a predefined structure and does not necessarily fit into rows and columns. It can include text, images, audio, video, or other formats that do not conform to a table-like structure. Typically, ML models such as CNN and GAN are used to handle these types of inputs.

- **Is the use of tabular/structured data the only reason why GBT outperforms MLP in this study? Is it possible that the size of the data**

**samples limits the capability of MLP, given that it is a deep learning technique after all?**

The use of tabular data could be one of the reasons for the better performance of GBT compared to MLP. The GBT algorithm is known for its ability to capture feature interactions effectively, which can be particularly advantageous when dealing with tabular data. On the other hand, the MLP algorithm might require a larger number of hidden layers and neurons to achieve similar performance. Additionally, the performance of MLP can also be affected by the sample size. Deep learning algorithms, including MLP, are known to be data-hungry and often require a large amount of data to generalize well. We have included a discussion on the sample size and other neural network algorithms in the revised manuscript.

| Line 231-235 | It is important to note that deep learning models are data-intensive, **and their performance and generalization capabilities tend to improve with larger amounts of data.** In our study, we utilized the simplest deep learning algorithm known as MLP. However, it is essential to explore the capabilities of other deep learning algorithms, such as CNN and LSTM, in future studies to gain further insights. Additionally, employing multiple ML models through bagging techniques could potentially lead to improved performance, despite the computational expense involved. |
| --- | --- |

- **In addition to the work of Heaton and Lundberg et al, can the authors find any other studies that have focused on the prediction of air pollutants that can support the results of this study?**

  There have been numerous studies conducted on deep learning models (Chan et. al., 2021) and traditional machine learning models (Zhu et. al., 2022) like Random Forest, XGBoost, etc., individually, to model air pollutant concentrations. However, we have not come across any studies that support our findings regarding the comparison between deep learning and tree-based models.

- **What about other neural network techniques? The author may not need to try them, but at least give a brief discussion, as MLP is one of the simplest deep learning algorithms.**

Thanks for the suggestion. In the revised manuscript, we have included a discussion on the sample size and other neural network algorithms in the revised manuscript, as given below.

| Line 231-235 | It is important to note that deep learning models are data-intensive, **and their performance and generalization capabilities tend to improve with larger amounts of data**. In our study, we utilized the simplest deep learning algorithm known as MLP. However, it is essential to explore the capabilities of other deep learning algorithms, such as CNN and LSTM, in future studies to gain further insights. Additionally, employing multiple ML models through bagging techniques could potentially lead to improved performance, despite the computational expense involved. |
|---|---|

**Section 3.3: In this section, the authors discuss the exceedances of NO2 and O3 using data produced by the GBT model, but the model's ability to capture extreme pollution is hardly evaluated in the validation section above. In fact, the scatter plot of Figure 4 indicates model does have a weakness in reproducing large NO2/O3 values. Therefore, I would suggest that the authors add this uncertainty discussion when analyzing people living beyond the WHO limit.**

Thanks for the suggestion. We agree with the reviewer that our model has some difficulty in capturing the extreme pollution events, as shown in figure 4. In order to evaluate the model capability in capturing the exceedance events (above WHO limit), we used the time-leave-out evaluation strategy. This approach is chosen because comparing the ML model simulations (after training with 100 % of data) with ground-truth is questionable as it was already used during the training process. In time-leave-out strategy, the exceedances of $NO_2$ and $O_3$ values simulated by GBT model are compared with Ground-truth exceedance events in each iteration. This allows us to assess the model's ability to reproduce the exceedance data that has not been used in the training process.

In both the $NO_2$ and $O_3$ GBT models, 82% of the WHO $NO_2$ and $O_3$ exceedance data in the whole dataset (Ground-truth) were correctly identified as WHO $NO_2$ and $O_3$ exceedance (True Positives), meaning 18% of actual WHO exceedances have not been identified as such by our GBT models (False Negatives). However, we also noted that 6.6% and 7.3% (False Positives) of the whole data were incorrectly identified as exceedance data by our $NO_2$ and $O_3$ GBT models, respectively (Table A6).

This discussion and table are included in the revised manuscript, as given below.

| Line 269-276 | We also evaluated the model capability in capturing the exceedance events (above WHO limit) using time-leave-out evaluation strategy. The exceedances of $NO_2$ and $O_3$ events simulated by GBT model compared with Ground-truth events in each iteration. This allows us to assess the model's ability to reproduce the exceedance events that have not been used in the training process. The 82% of the WHO $NO_2$ and $O_3$ exceedance events in the whole dataset (Ground-truth) were correctly identified as WHO $NO_2$ and $O_3$ exceedance events (True Positives) in both the $NO_2$ and $O_3$ GBT models (Table A5). However, we also noted that 6.6% and 7.3% of the whole data were incorrectly identified as exceedance events by our $NO_2$ and $O_3$ GBT models, respectively (False Positives). This indicates that our GBT model might slightly underestimate the exceedance events for both $NO_2$ and $O_3$. This could be due to unknown drivers that are not included in the model. |
|---|---|

Table A6. Comparison between WHO $NO_2$ and $O_3$ exceedance events in the ground-truth dataset and GBT simulated WHO $NO_2$ and $O_3$ exceedance events using time-leave-out testing strategy.

| | **Ground-truth WHO exceedance** | **Correct detection as exceedance by GBT model (True Positives)** | **Incorrect detection as exceedance by GBT model (False Positives)** |
|---|---|---|---|
| **Near-surface $NO_2$** | 36772 | 30125 | 7439 |
| **Near-surface $O_3$** | 35860 | 29396 | 6924 |

**In addition, a temporal evaluation of the daily time-series (CAMS/GBT versus ground-truth O3) may be meaningful, such as using the temporal correlation coefficient.**

As discussed above, it is questionable to compare the ground-truth $O_3$ values to the model predictions (after training with 100 % of ground-truth data). This is because the model is fitted based on the ground-truth $O_3$. However, we compared CAMS vs Ground-truth and GBT vs Ground-truth for the period between 17-07-2019 and

31-01-2020 (this time period was not used for training the GBT model for this comparison). This evaluation strategy involves comparing the model predictions with the ground-truth $O_3$ for a particular period, which is not included in the training dataset (Figure 4). The outcome of this evaluation, along with the results of the time-leave-out evaluation strategy results, provides valuable insight into the model's temporal correlation coefficient.

[Figure]

Figure 4. Top: Comparison between ground-truth near-surface $NO_2$ and CAMS reanalysis near-surface $NO_2$ (a) and $O_3$ (b) for the period between 17-07-2019 and 31-01-2020. Bottom: Comparison between ground-truth near-surface $NO_2$ and GBT-simulated near-surface $NO_2$ (c) and $O_3$ (d) for the period between 17-07-2019 and 31-01-2020. The dotted line represents a 1:1 line, while the solid line represents a linear fit.

**References:**

Chan, K. L., Khorsandi, E., Liu, S., Baier, F., and Valks, P.: Estimation of surface NO2 concentrations over Germany from TROPOMI satellite observations using a machine learning method, Remote Sensing, 13, 969, 2021.

Zhu, Q., Bi, J., Liu, X., Li, S., Wang, W., Zhao, Y., and Liu, Y.: Satellite-Based Long-Term Spatiotemporal Patterns of Surface Ozone Concentrations in China: 2005–2019, Environmental health perspectives, 130, 027 004, 2022.